# A conserved regulatory mechanism mediates the convergent evolution of plant shoot lateral organs

Satoshi Naramoto[1]*, Victor Arnold Shivas Jones[2], Nicola Trozzi[1,3], Mayuko Sato[4], Kiminori Toyooka[4], Masaki Shimamura[5], Sakiko Ishida[6], Kazuhiko Nishitani[7], Kimitsune Ishizaki[8], Ryuichi Nishihama[6], Takayuki Kohchi[6], Liam Dolan[2], Junko Kyozuka[1]

1 Graduate School of Life Sciences, Tohoku University, Sendai, Japan, 2 Department of Plant Sciences, University of Oxford, Oxford, United Kingdom, 3 Department of Molecular Biology, Umeå University, Umeå, Sweden, 4 RIKEN Center for Sustainable Resource Science, Yokohama, Japan, 5 Graduate School of Integrated Sciences for life, Hiroshima University, Higashi-Hiroshima, Japan, 6 Graduate School of Biostudies, Kyoto University, Kyoto, Japan, 7 Department of Biological Sciences, Kanagawa University, Hiratsuka, Japan, 8 Graduate School of Science, Kobe University, Kobe, Japan

* satoshi.naramoto.d6@tohoku.ac.jp

**Data Availability Statement:** All relevant data are within the paper and its Supporting Information files.

## Abstract

Land plant shoot structures evolved a diversity of lateral organs as morphological adaptations to the terrestrial environment, with lateral organs arising independently in different lineages. Vascular plants and bryophytes (basally diverging land plants) develop lateral organs from meristems of sporophytes and gametophytes, respectively. Understanding the mechanisms of lateral organ development among divergent plant lineages is crucial for understanding the evolutionary process of morphological diversification of land plants. However, our current knowledge of lateral organ differentiation mechanisms comes almost entirely from studies of seed plants, and thus, it remains unclear how these lateral structures evolved and whether common regulatory mechanisms control the development of analogous lateral organs. Here, we performed a mutant screen in the liverwort *Marchantia polymorpha*, a bryophyte, which produces gametophyte axes with nonphotosynthetic scalelike lateral organs. We found that an *Arabidopsis* LIGHT-DEPENDENT SHORT HYPOCOTYLS 1 and *Oryza* G1 (ALOG) family protein, named *M. polymorpha* LATERAL ORGAN SUPRESSOR 1 (MpLOS1), regulates meristem maintenance and lateral organ development in *Marchantia*. A mutation in Mp*LOS1*, preferentially expressed in lateral organs, induces lateral organs with misspecified identity and increased cell number and, furthermore, causes defects in apical meristem maintenance. Remarkably, Mp*LOS1* expression rescued the elongated spikelet phenotype of a Mp*LOS1* homolog in rice. This suggests that ALOG genes regulate the development of lateral organs in both gametophyte and sporophyte shoots by repressing cell divisions. We propose that the recruitment of ALOG-mediated growth repression was in part responsible for the convergent evolution of independently evolved lateral organs among highly divergent plant lineages, contributing to the morphological diversification of land plants.

**Funding:** The funders of this research are the Grants-in-Aid from the Ministry of Education, Culture, Sports and Technology, Japan (https://www.jsps.go.jp/english/e-grants/) (KAKENHI grant numbers 17K17595 for SN, 18H04836 for RN, 17H06472 for KI, and 17H06465 for JK); Newton Abraham Studentship from the University of Oxford, to VASJ; and ERC advanced Grant EVO-500 (250284) (https://erc.europa.eu/funding/advanced-grants) to LD. The funders had no role in study design, data collection and analysis, decision to publish, or preparation of the manuscript.

**Competing interests:** The authors have declared that no competing interests exist.

**Abbreviations:** *35Spro*, *cauliflower mosaic virus 35S promoter*; ALOG, *Arabidopsis* LSH1 and *Oryza* G1; AtLSH3, *Arabidopsis thaliana* LSH3; EdU, 5-ethynyl-2′-deoxyuridine; eGFP, enhanced GFP; *G1*, long sterile lemma; FESEM, field emission SEM; GFP, green fluorescent protein; GUS, β-glucuronidase; LOS1, LATERAL ORGAN SUPRESSOR 1; LSFM, light sheet fluorescence microscopy; LSH, LIGHT-DEPENDENT SHORT HYPOCOTYLS; LTI6B, LOW TEMPERATURE INDUCED PROTEIN 6B; MpLOS1, *M. polymorpha* LOS1; OsTAW1, *Oryza sativa* TAWAWA1; *proMpEF*, promoter *M. polymorpha elongation factor-1 alpha*; *proMpYUC2*, promoter *M. polymorpha YUCCA2*; SEM, scanning electron microscope; SlTMF, *Solanum lycopersicum* TERMINATING FLOWER; T-DNA, transfer DNA; WT, wild type.

## Introduction

During 470 million years of evolution, the body plans of land plants diversified independently among the gametophyte and sporophyte life stages of different plant groups. In extant bryophytes, early-diverging land plants, the gametophyte is the dominant phase of the life cycle [1,2]. The gametophyte comprises an apical–basal axis with an apical stem cell and forms structures in which gametes develop (antheridiophores and archegoniophores). In contrast, the sporophyte is dominant in extant vascular plants. The sporophyte comprises an axial system (shoots or stems growing along apical–basal axes) that develops from an apical meristem and forms structures in which haploid spores develop. Therefore, in different plant lineages, gametophytes and sporophytes develop axial systems that are produced by apical meristems [3–5].

Extant bryophytes and vascular plants develop lateral organs on gametophytes and sporophytes, respectively. Apical meristems maintain stem cell activity at their center and iteratively generate lateral organs at the meristem periphery. The spatiotemporal differences in cell division and expansion in lateral organs contribute to the morphological diversity of shoot structures in land plants [6–11].

The liverwort *Marchantia polymorpha* is a bryophyte that forms a gametophyte axis that undergoes indeterminate planar growth in the form of a flattened mat of tissue, called a thallus. The thallus exhibits strong dorsoventrality; gemma cups, gemmae, and air chambers develop on the dorsal side, whereas rhizoids and ventral scales are formed on the ventral side (Fig 1A) [12–17]. Ventral scales cover bundles of rhizoids that run along the underside of the thallus and facilitate water and nutrient transport over the ventral surface of the thallus (Fig 1B) [17]. In the leafy liverworts, photosynthetic leaves arise next to a tetrahedral single stem cell (apical cell). By contrast, *M. polymorpha* does not develop photosynthetic leaves. Instead, the ventral scales alternately develop on the left and right sides of the wedge-shaped apical cell on the ventral surface in the apical notch near the growing tip of the thallus. The flattened form, single-cell thickness, and bilateral symmetry of the *Marchantia* scales resemble the leaves in leafy liverworts (Fig 1C) [16,17]. The ventral scales of *M. polymorpha* are hypothesized to be homologous to the photosynthetic leaves of the basally diverging leafy liverworts [18,19].

The fossil record indicates that the shoot of the earliest known land plants comprised branching stems without lateral, determinate organs [20,21]. Subsequently, determinate lateral organs, which develop from the sides of apical meristems, evolved. The earliest example of such a lateral organ is the microphyll that developed on the stems of the sporophyte of *Baragwanathia longifolia*, a lycophyte, which first appears in the fossil record in the late Silurian [22,23]. No lateral organs are known from the gametophytes of early bryophytes from the Silurian or Devonian; these likely arose subsequently and are found in extant bryophytes. The acquisition and modification of different lateral organ types are likely to have been morphological adaptations to the terrestrial environment that increased photosynthetic efficiency, gas exchange, and water transport [7,24–26].

Mechanisms controlling lateral organ development are well described in angiosperms such as rice and *Arabidopsis*. However, little is known about the mechanisms that regulate lateral organ development among bryophytes. Therefore, we carried out a forward genetic screen for mutants with defective lateral organ development in the liverwort *M. polymorpha* to define mechanisms that control lateral organ development in this species. Comparing the roles of the genes that control lateral organ development in liverworts and angiosperms allows the identification of mechanisms that were involved in the independent evolution of analogous lateral organs during land plant evolution [7,24–26].

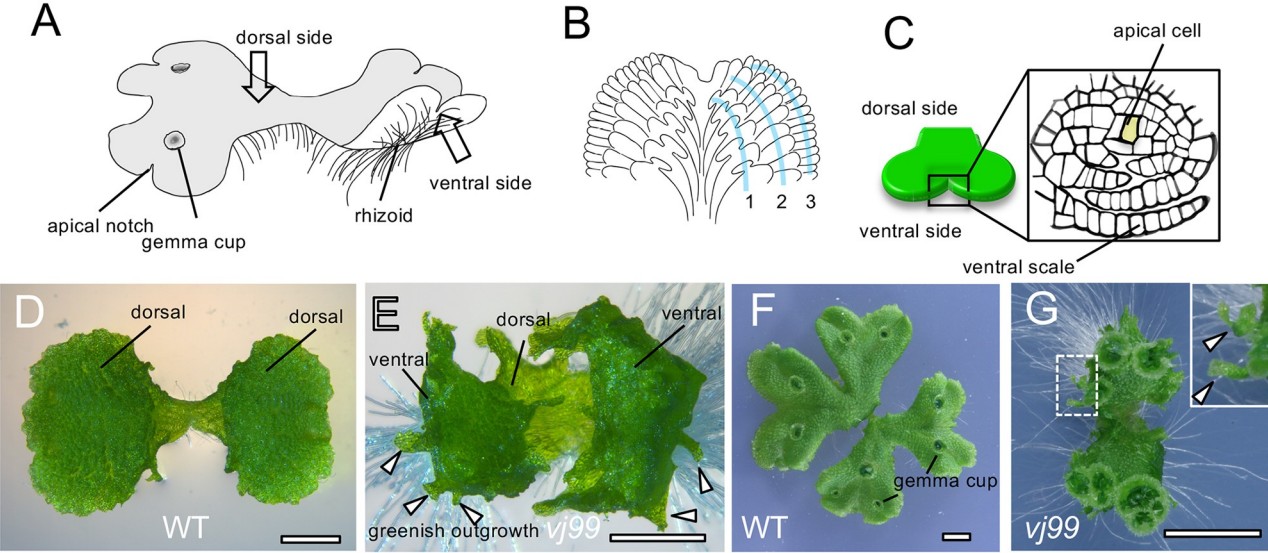

**Fig 1. Illustration of *M. polymorpha* thallus and the isolation of Mp*los1* mutants.** (A-C) Diagrammatic representation of vegetative *M. polymorpha* thallus. Gross morphology (A), ventral side of thallus with ventral scales arranged in three rows on each side of the thallus (B), and vertical transverse section of a notch region (C). Rhizoids are not shown in (B) to clearly visualize ventral scales. The apical cell is shaded in (C). (D-G) Gross morphology of WT Tak1 (D and F) and Mp*los1-1* mutant *vj99* (E and G) gemmalings. The 10-day-old gemmalings (D and E) and 3-week-old (F and G) thalli are shown. The dotted square box in (G) that includes the abnormal green outgrowth is enlarged in the image in the top-right corner. Arrowheads indicate abnormal green outgrowths. Note that, unlike WT, Mp*los1-1* mutants displayed upward bending of thalli and formed green outgrowths. Scale bars = 1 mm in (D and E) and 0.5 cm in (F and G). Mp*los1*, *M. polymorpha LATERAL ORGAN SUPPRESSOR 1*; Tak1, Takaragaike-1; WT, wild type.

## Results

### *M. polymorpha* LATERAL ORGAN SUPPRESSOR 1 specifies lateral organ identity during vegetative growth

We isolated two mutants, *vj99* and *vj86*, that produced abnormal green outgrowths from a population of 105,000 transfer DNA (T-DNA)-transformed *M. polymorpha* (Fig 1D–1G; S1A and S1B Fig). *vj99* and *vj86* thalli were hyponastic, bending upward at the thallus margins, unlike wild type (WT) (Fig 1D and 1E; S1C and S1D Fig). A single T-DNA was inserted into the gene *Mapoly0028s0118* in *vj99* and *vj86*, suggesting that defective function of *Mapoly0028s0118* was responsible for the green outgrowths (S1E Fig). To test this hypothesis, we generated independent mutations in the *Mapoly0028s0118* gene by homologous recombination. Mutants of *Mapoly0028s0118* generated by targeted deletion developed similar phenotypes to those of the *vj99* and *vj86* mutants (S1F and S1G Fig). To verify that a defect in *Mapoly0028s0118* was responsible for the green outgrowth, we transformed mutant *vj99* with a genomic fragment that includes the full-length *Mapoly0028s0118* gene. Transformation of the *Mapoly0028s0118* genomic fragment into *vj99* mutants restored WT development, demonstrating that loss of *Mapoly0028s0118* function causes the *vj99* phenotype (S1H–S1K Fig). Phylogenetic analysis indicated that *Mapoly0028s0118* belongs to the *Arabidopsis* LIGHT-DEPENDENT SHORT HYPOCOTYLS 1 (LSH1) and *Oryza* G1 (ALOG) protein family (S1L Fig). The proteins in this family contain a DNA-binding domain with weak transcriptional activity [27–29]. We named this gene *M. polymorpha LATERAL ORGAN SUPPRESSOR 1* (Mp*LOS1*). In addition to the abnormal green outgrowths, gemma cup spacing was abnormal in the Mp*los1-1* (*vj99*) mutant; the distance between neighboring gemma cups was much shorter than in the WT (Fig 1F and 1G).

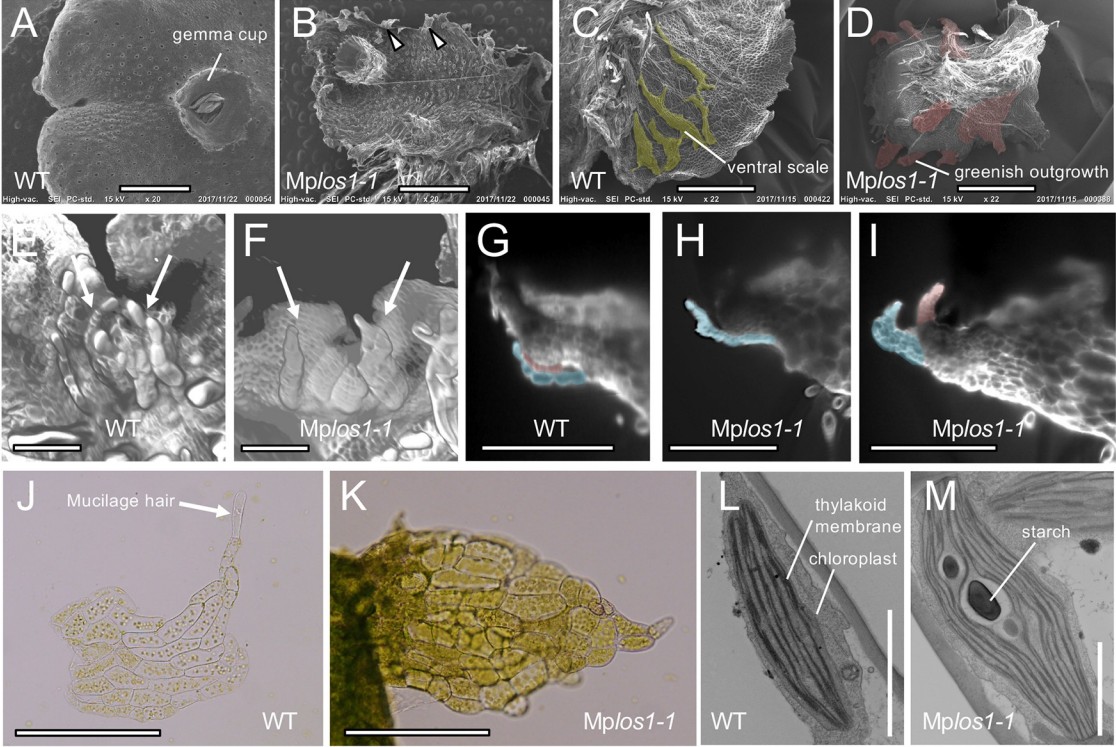

**Fig 2. MpLOS1 functions are required for specification of lateral organs.** (A-D) SEM images of WT (A and C) and Mp*los1-1* mutant (B and D) thalli. Dorsal (A and B) and ventral (C and D) sides of thalli are shown. Ventral scales in (C) and greenish outgrowth in (D) are highlighted in yellow and red, respectively. (E-I) LSFM images of WT (E and G) and Mp*los1-1* mutants (F, H, and I). Observation of apical notch regions from the ventral side (E and F) revealed the role of MpLOS1 in specifying lateral organs as ventral scales (F). Vertical longitudinal optical sections (G-I) around apical notches identified accelerated cell division of lateral organs in Mp*los1-1* mutants (H and I). Lateral organs are indicated by arrows in (E and F) and highlighted in red or blue in (G-I). (J and K) Images of ventral scales in WT (J) and the corresponding tissues in Mp*los1-1* mutants (K). Note that ventral scale cells are transformed into green tissues that lack mucilage hair cells in Mp*los1-1* mutants. Mucilage hair in WT is indicated by an arrow in (J). (L and M) FESEM images of ventral scale cells in WT (L) and the corresponding cells in Mp*los1-1* mutants (M). Three independent gemmalings were analyzed in WT and in Mp*los1-1* mutants, and in total, 97 and 262 chloroplasts were analyzed in WT and in Mp*los1-1* mutants, respectively. Note that ventral scale cells are transformed into photosynthetic cells in Mp*los1-1* mutants. Scale bars = 1.5 mm in (A-D), 150 μm in (E and F), 300 μm in (G-I), 200 μm in (J and K), and 2 μm in (L and M). LSFM, light sheet fluorescence microscopy; MpLOS1, *M. polymorpha* LATERAL ORGAN SUPRESSOR 1; FESEM, field emission scanning electron microscope; WT, wild type.

To more precisely define the nature of the green outgrowths, we performed a phenotypic analysis of the Mp*los1-1* mutant. Outgrowths emerged from the ventral side of the thallus near the thallus margins and extended beyond the thallus margin in the Mp*los1-1* mutant (Fig 2A–2D). These outgrowths resembled ventral scales in a number of ways. They developed in pairs near the apical notch (Fig 2E and 2F; S2A and S2B Fig). They were in general a single cell layer thick, although the outgrowths located near the apical notch occasionally consist of several cell layers (Fig 2G–2I). Outgrowths located near the apical notch also tended to pile up on one another at the edge of the ventral surface (Fig 2G and 2I). Furthermore, although outgrowths developed, no ventral scales formed on Mp*los1-1* mutants (Fig 2C and 2D), suggesting that the outgrowths are modified ventral scales. Taken together, these data suggested that the outgrowths formed from the ventral thallus on Mp*los1-1* mutants were related to ventral scales.

Although similar to ventral scales, these outgrowths differ in a number of characteristics. The abnormal outgrowths formed in Mp*los1-1* mutants were greener than typical ventral scales (Fig 2J and 2K). There were more cells in outgrowths than in WT scales (Fig 2G–2I; S2C–S2F Fig). Moreover, the mutant chloroplasts were larger than in WT, and there were more thylakoid membranes in the mutants than in WT (Fig 2L and 2M; S2I and S2J Fig). Rhizoids never differentiated in the green outgrowths of the Mp*los1-1* mutants, unlike those in WT ventral scales (S2G and S2H Fig). Taken together, these observations indicated that MpLOS1 plays crucial roles in specifying lateral organs as ventral scales, in which MpLOS1 inhibits cell division and chloroplast differentiation.

## MpLOS1 activity is required for the maintenance of meristem activity

The WT thallus comprises an apical–basal axis produced by the activity of apical stem cells. The thallus undergoes periodic bifurcation, and gemma cups develop along the midline of the dorsal surface. When the WT thallus bifurcates, a notch containing an apical stem cell forms on each of the two new apical–basal axes (Fig 3A). This process, the duplication of apical notches and the subsequent growth of thalli, is termed "branching." Upon branching, adjacent apical notches are initially pushed away by the growth of tongue-like tissues, called central lobes, and subsequently further separated concomitant with the growth of the thallus (Fig 3A) [30]. Gemma cups initiate from dorsal merophytes, clones derived from the cell that are cut off from the dorsal surface of the apical cell [17], and they are regularly spaced along the dorsal midline of each thallus.

Gemma cups were more densely arranged along the dorsal surface of Mp*los1-1* mutants than in WT (Fig 1G). This suggested that the mutants had defects in gemma cup differentiation or axis development, or both. To address whether Mp*LOS1* is involved in bifurcation or gemma cup differentiation, we analyzed the number of apices and gemma cups in Mp*los1-1* mutants during cultivation. To count meristems, we imaged expression of the *promoter M. polymorpha YUCCA2*: *ß-glucuronidase* (*pro*Mp*YUC2*:*GUS*) construct, which is preferentially expressed in notches [31]. The number of apical notches expressing GUS was not significantly different between WT and Mp*los1-1* mutants until day 7 of cultivation, although subsequently, fewer GUS-expressing apical notches were detected in the Mp*los1-1* mutants compared with WT (Fig 3B–3D). This indicates that bifurcation occurs normally, at least in the early stages of development. In contrast, the density of apical notches in Mp*los1-1* mutants was higher than WT at 3 weeks (S3A and S3B Fig). Importantly, there is no clear difference in the number of gemma cups between WT and Mp*los1-1* mutants at day 17 of cultivation, although subsequently, fewer gemma cups with higher density were formed in Mp*los1-1* mutants (S3B and S3C Fig). These data suggested that the onset of bifurcation, as well as gemma cup differentiation, is not affected and that the separation process of each apical notch is compromised in Mp*los1-1* mutants. The lower number of GUS-positive notches and gemma cups in Mp*los1-1* mutants after prolonged cultivation may be due to a secondary effect of slow thallus growth or technical limitations of counting densely clustered apices and small-sized immature gemma cups.

The separation of apical notches is dependent on the division and expansion of cells between notches, where central lobes develop (Fig 3A). We reasoned that defective cell division in apical notches and/or central lobes in Mp*los1-1* mutant thalli would lead to defects in apical notch separation. We analyzed the cell division activity of Mp*los1-1* mutant gemmalings during 7 days' cultivation by applying a 3-hour pulse of 5-ethynyl-2′-deoxyuridine (EdU), a thymidine analog that is incorporated into cells during DNA replication (Fig 3E–3G) [32]. In early development (days 1–3), the number of cells labeled by EdU was indistinguishable

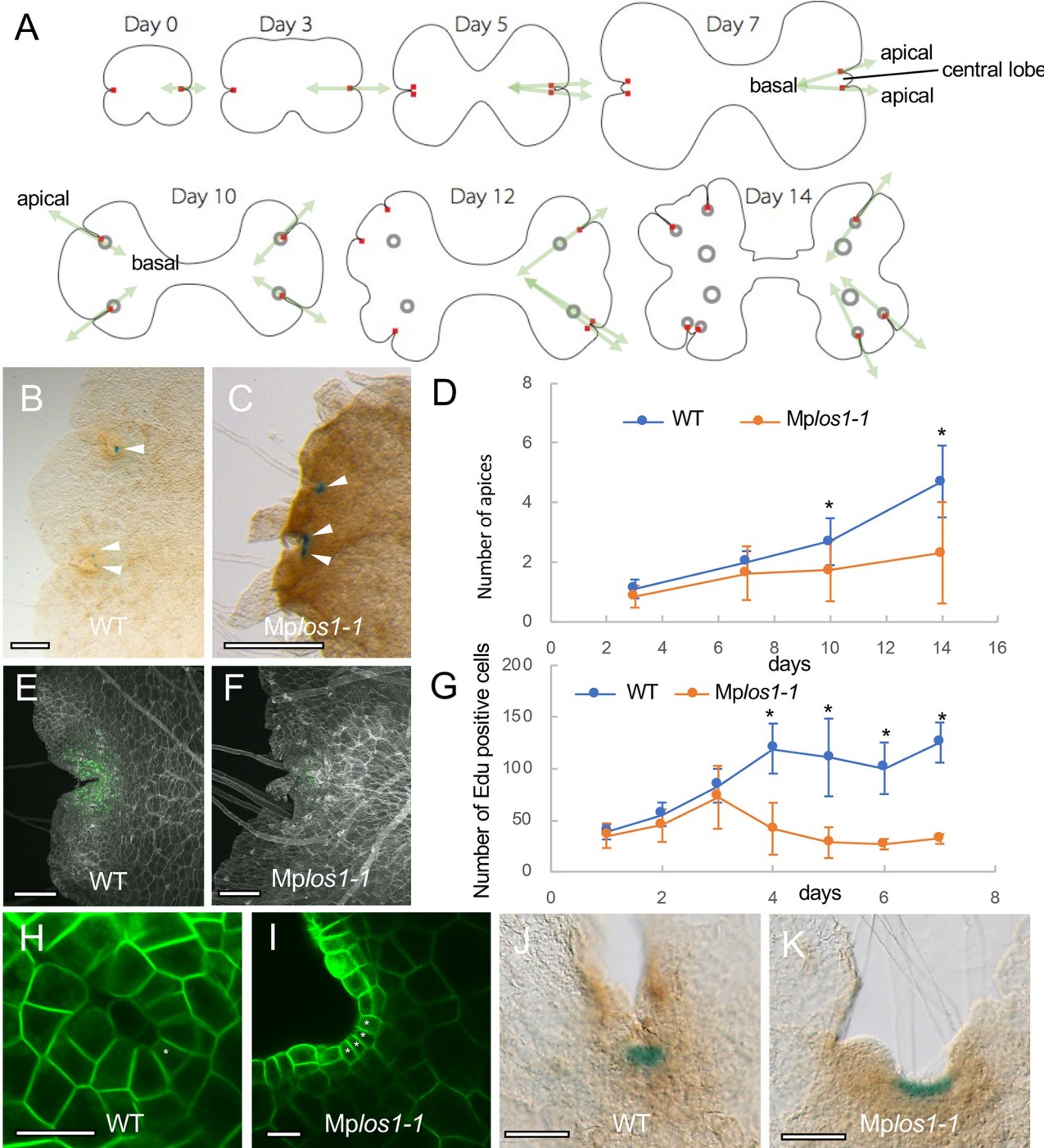

**Fig 3. Loss-of-function mutant of MpLOS1 displays defects in meristem maintenance.** (A) Diagrammatic representation of thallus shape transition in *M. polymorpha* thallus development. Note that distances between duplicated apical notches as well as gemma cups are gradually increased along with the progression of branching. Red squares and black circles indicate apical cells and gemma cups, respectively. Apical–basal axes formed in thalli are indicated by green arrows. (B and C) Apices stained by *pro*Mp*YUC2*:*GUS* in 10-day-old gemmalings in WT (Tak1) (B) and Mp*los1-1* mutants (C). Arrowheads indicate GUS staining at apical notches. (D) The number of apices stained by *pro*Mp*YUC2*:*GUS* in Mp*los1-1* mutants as compared with Tak1 throughout 14 days of gemmaling growth. The structure of a gemma is symmetrical, with a single notch on either side. The number of apices originating from one side of each gemma (hereafter referred to as a "half gemmaling") was counted. Each time point indicates the mean ± SD. At least 14 gemmalings were analyzed at each time point. *p*-Values lower than 0.01 were indicated by asterisks (*). (E-G) Cell division activities in Mp*los1-1* mutants decreased after 3 days' incubation. EdU-positive signals of 4-day-old gemmalings in Tak1 (E) and Mp*los1-1* mutants (F) are shown in green. (G) EdU uptake activities of Tak1 and Mp*los1-1* mutants at the indicated time points. EdU-positive signals detected within half gemmalings were

counted. Each time point indicates the mean ± SD. At least six gemmalings were analyzed at each time point. *p*-Values lower than 0.01 are indicated by asterisks (*). (H-K) Defective apical meristem structures in Mp*los1-1* mutants. A single apical cell, as seen in 3-day-old Tak1 gemmalings (H), was not observed in Mp*los1-1* mutants (I). GUS staining of *pro*Mp*YUC2:GUS* gemmalings in the apical notch region was broader in Mp*los1-1* mutants (K) as compared with Tak1 (J). PMs in (H) and (I) were labeled by *pro*Mp*EF:LTI6B-GFP* constructs. Asterisks indicate wedge-shaped cells that are either apical cells or lateral merophytes. Scale bars = 500 μm in (B and C), 200 μm in (E and F), 20 μm in (H and I), and 100 μm in (J and K). Underlying data for this figure can be found in S2 Data. EdU, 5-ethynyl-2′-deoxyuridine; GFP, green fluorescent protein; GUS, β-glucuronidase; *LTI6B*, *LOW TEMPERATURE INDUCED PROTEIN 6B*; MpLOS1, *M. polymorpha* LATERAL ORGAN SUPRESSOR 1; PM, plasma membrane; *pro*Mp*EF*; promoter *M. polymorpha* elongation factor-1 alpha; *pro*Mp*YUC2*, promoter *M. polymorpha* YUCCA2; Tak1, Takaragaike-1; WT, wild type.

between the WT and Mp*los1-1* mutants (Fig 3G). However, beginning from day 3, the incorporation of EdU was lower in Mp*los1-1* mutants than in WT, which suggested that cell division is reduced in Mp*los1-1* mutants compared with WT (Fig 3E–3G). To determine whether the cell divisions that led to the separation of the apical notches during branching occur in the apical notches themselves or within the central lobes after they emerge, we measured EdU incorporation in these areas during WT development. We found that many cells incorporated EdU in the incipient central lobe as it forms in the apical notch region (S3D Fig). In contrast, very little cell division is found within the central lobe once it has emerged (S3E Fig). This indicates that cells in central lobes are mainly supplied from the apical notches, whereas the contribution of cell division activity within the central lobe for separating duplicated apical notches is low. Altogether, these data suggest that the rate of cell division in the apical notch is lower in the mutant than in WT, leading to an impaired separation of apical notches during branching.

We also compared the cellular organization of apical meristems of Mp*los1-1* mutants and WT. The WT apical meristem comprised a single wedge-shaped apical cell and surrounding merophytes (a group of clonally related cells resulting from sequential cell divisions in a single derivative of the apical cell of a meristem), in which the lateral merophytes and the apical cell display identical shapes (Fig 3H) [17]. In contrast, there were many wedge-shaped cells in Mp*los1-1* mutants, in contrast to the single apical cell of WT (Fig 3I; S4A–S4H Fig). These defective cell division patterns in Mp*los1-1* mutants led to the formation of less strongly invaginated notch regions (Fig 3I; S4A–S4H Fig). Whereas the expression of *pro*Mp*YUC2:GUS* was restricted to a small area of the WT apical notch, staining was more dispersed in Mp*los1-1* mutants (Fig 3J and 3K). Occasionally (3 out of 20 gemmalings at 14 days' cultivation), apical meristems were aborted in Mp*los1-1* mutants (S4I Fig, dotted boxes), a phenomenon not observed in WT in our conditions. These data suggest that MpLOS1 is required for the maintenance of apical meristems. These data also support the hypothesis that Mp*los1-1* mutants fail to separate apical notches because of defects in cell proliferation in the apical notches. Gemma cup differentiation as well as bifurcation initiate as in WT, but then subsequent defective meristem activity causes defective axis expansion, resulting in the development of a higher density of gemma cups and apical notches in the Mp*los1-1* mutant thallus.

## MpLOS1 is expressed in lateral organs but not in apical cells

To define the spatial expression patterns of Mp*LOS1*, we established a line that expressed GUS under the control of 5′ and 3′ regulatory elements that were used in the complementation analysis of Mp*los1-1* mutants (S1E Fig). In 4-day-old gemmalings, GUS staining was detected in notches and rhizoids (Fig 4A). Weak signal was observed elsewhere in growing thalli (Fig 4B and 4C). The developing ventral scales in the ventral region of the apical notch stained the strongest (Fig 4B–4F). Staining extended over the entire young ventral scale and the basal region of old ventral scales (Fig 4D and 4E). No signal was detected in the oldest ventral scales (Fig 4D and 4E). The expression of Mp*LOS1* in ventral scales is consistent with the phenotypic defects seen in these organs in the mutant, further strengthening our hypothesis that Mp*LOS1*

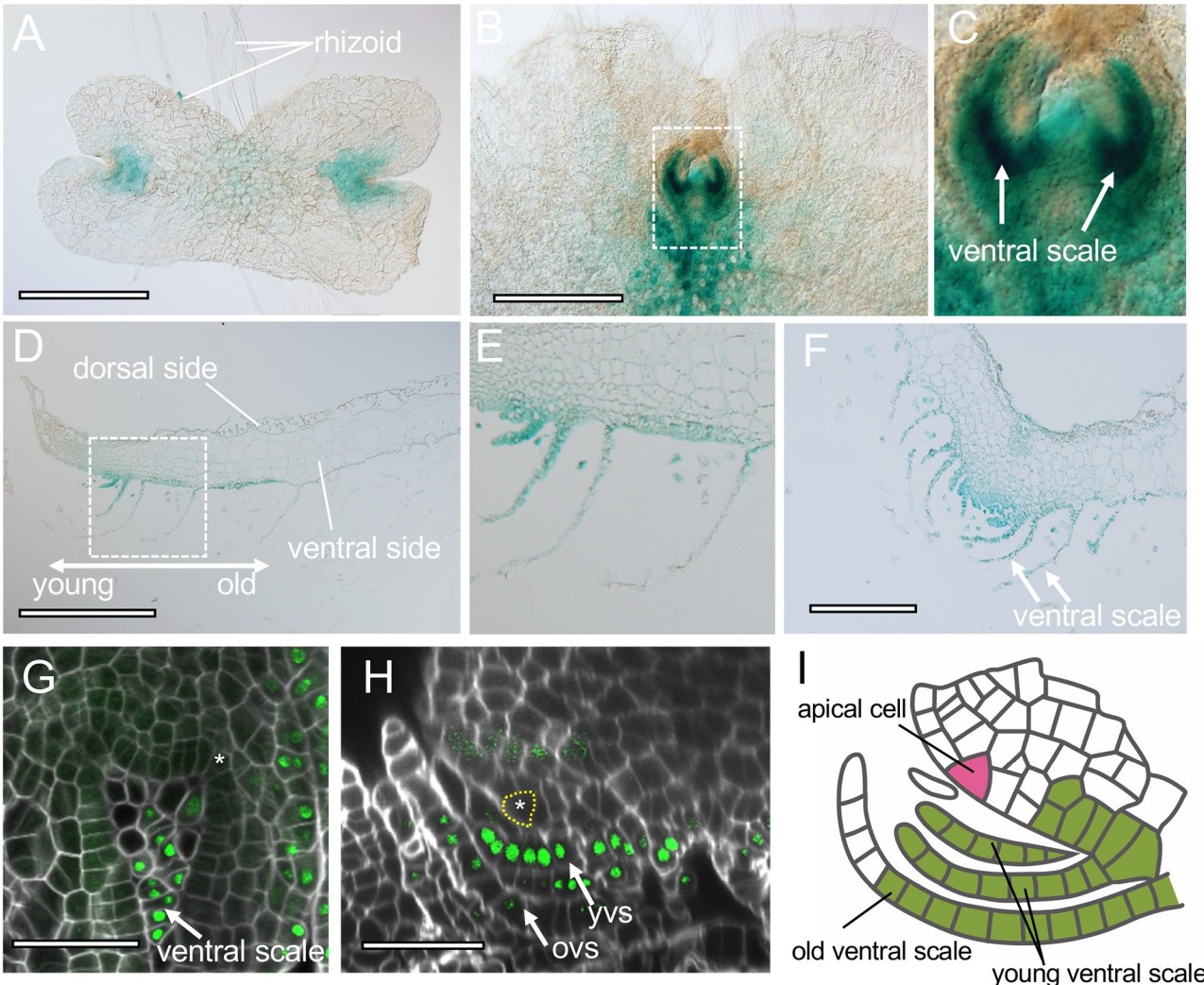

**Fig 4. MpLOS1 regulates lateral organ development cell autonomously and meristem maintenance non–cell autonomously.** (A-C) GUS activity in ventral thalli of 4-day-old (A) and 10-day-old (B and C) *pro*Mp*LOS1*:*GUS*-expressing gemmalings. (C) is a close-up image of the dotted square depicted in (B). Images were taken from the ventral side. (D-F) Cross section of GUS-stained 10-day-old *pro*Mp*LOS1*:*GUS* gemmalings. (D and E) are vertical longitudinal sections, and (F) is a vertical transverse section. (E) is a close-up image of the dotted square depicted in (D). "Young" and "old" in (D) indicates the relative age of ventral scales. (G and H) Functional eGFP-MpLOS1 proteins were not detected in apical meristems but were detected in ventral scales and the cells beneath the basal part of ventral scales. Confocal images of a horizontal optical section (G) and a vertical longitudinal optical section (H) of Mp*los1-1* mutant gemmalings that express *pro*Mp*LOS1*:*eGFP*-Mp*LOS1* constructs are shown. Cell walls were stained by calcofluor. Apical cells are indicated by asterisks and/or dotted yellow lines. (I) Schematic of MpLOS1 protein localization around apical notches. MpLOS1 protein is detected in ventral scales and the cells located around the basal part of ventral scales. Note that MpLOS1 protein is not present in apical cells. Green indicates cells that contain MpLOS1 protein. The apical cell is in pink. Scale bars = 500 μm in (A, B, and D), 200 μm in (F), and 50 μm in (G and H). GFP, green fluorescent protein; GUS, ß-glucuronidase; MpLOS1, *M. polymorpha* LOS1; ovs, old ventral scale; *pro*Mp*LOS1*, promoter *M. polymorpha LOS1*; yvs, young ventral scale.

is required for the normal development of ventral scales. We also expressed functional *pro*Mp*LOS1*:*enhanced green fluorescent protein* (*eGFP*)-Mp*LOS1* constructs in Mp*los1-1* mutants (S1H, S1I and S1K Fig) to analyze the distribution of MpLOS1 protein on a cellular level. eGFP-MpLOS1 protein was preferentially detected in the ventral parts of the apical notch regions (Fig 4G and 4H; S5 Fig; S1 Movie). In particular, stronger signals were detected all over the young ventral scales as well as at the basal region of old ventral scales (Fig 4G and 4H; S5 Fig; S1 Movie). These results further supported a crucial role for MpLOS1 in the specification of lateral organs as ventral scales. However, eGFP-MpLOS1 proteins were not detected in

apical cells or lateral merophytes despite the defect in apical meristem morphology and maintenance in Mp*los1-1* mutants (Fig 4G–4I). These findings suggest that MpLOS1 mediates the maintenance of apical meristems non–cell autonomously, although we cannot exclude the possibility that MpLOS1 proteins below the level of detection in the apical meristems maintain meristem activity (Fig 4I).

## MpLOS1 specifies lateral organ identity during reproductive growth

*M. polymorpha* produces an umbrellalike gametangiophore (antheridiophore or archegoniophore) that bears antheridia or archegonia during reproductive growth [17]. The gametangiophore is a vertically growing thallus branch [17], and we reasoned that gametangiophore development might be defective in Mp*los1-1* mutants. The antheridial receptacles of male Mp*los1-1* plants were smaller than WT, and unlike in the WT, antheridia were frequently exposed (S6A–S6C Fig). Moreover, the scales on the antheridial receptacles of Mp*los1-1* plants were larger than WT (S6D–S6H Fig). Mp*LOS1* expression was detected in the ventral scales of the antheridiophore, as well as jacket cells, mucilage cells, and throughout the antheridia in plants harboring the *pro*Mp*LOS1*::*GUS* transgene (S6I and S6J Fig). These observations demonstrate that MpLOS1 regulates ventral scale development by restricting cell division in both the vegetative and reproductive phases.

The archegonial receptacle of female *M. polymorpha* is highly lobed, with finger-like structures called digitate rays (Fig 5A). The archegonial receptacle lacks the rows of typical ventral scales that develop in antheridiophores. Instead, a pair of specialized scalelike structures called involucres, which are larger than ventral scales, develop between each digitate ray and enclose the archegonia cluster (Fig 5C and 5D) [17]. In female Mp*los1-2* mutants (*vj86*), large leaf-like structures developed as in antheridiophores (Fig 5E). Moreover, more than two involucre-like structures differentiated between each digitate ray (Fig 5C–5F). Importantly, these involucre-like structures resembled ventral scales in their arrangement in several rows (Figs 1B, 1C, 5G and 5H). This suggests that loss of MpLOS1 function results in the transformation of involucres into more scalelike structures. GUS staining was also detected in immature involucres but not in mature involucres in *pro*Mp*LOS1*::*GUS* archegoniophores (Fig 5I; S6K Fig), accompanied by staining of all parts of the archegonia including eggs, collars, and venters (Fig 5I; S6K and S6L Fig) [17]. These data suggest that ventral scales are transformed into involucres in a MpLOS1-dependent manner upon the transition from vegetative to reproductive growth, in which MpLOS1 inhibits the growth of two rows of ventral scales, resulting in the formation of a single pair of involucres.

## Protein function of ALOG proteins are conserved between *Marchantia* and rice

Mutation of *long sterile lemma* (*G1*), a member of the *ALOG* gene family in rice, results in the enlargement of sterile lemmas, a small leafy lateral organ in the rice spikelet (flower clusters in grass species, basic unit of inflorescence, consisting of one or more flowers), which is interpreted as a homeotic transformation of a sterile lemma into a lemma [29]. Similarly, Mp*los1-2* mutants displayed defects in lateral organ specification that we interpret as the transformation of involucres into ventral scalelike structures. To determine whether the *M. polymorpha* protein could rescue the homeotic transformation of the rice mutant, we expressed Mp*LOS1* in rice *g1* mutants. Expression of Mp*LOS1* restored the WT short sterile lemma phenotype (Fig 6A–6C). This suggests that the protein functions of the ALOG proteins have been conserved since the time that *M. polymorpha* and rice last shared a common ancestor, which likely lacked lateral organs. It further suggests that ALOG family proteins were independently co-opted to specify

 

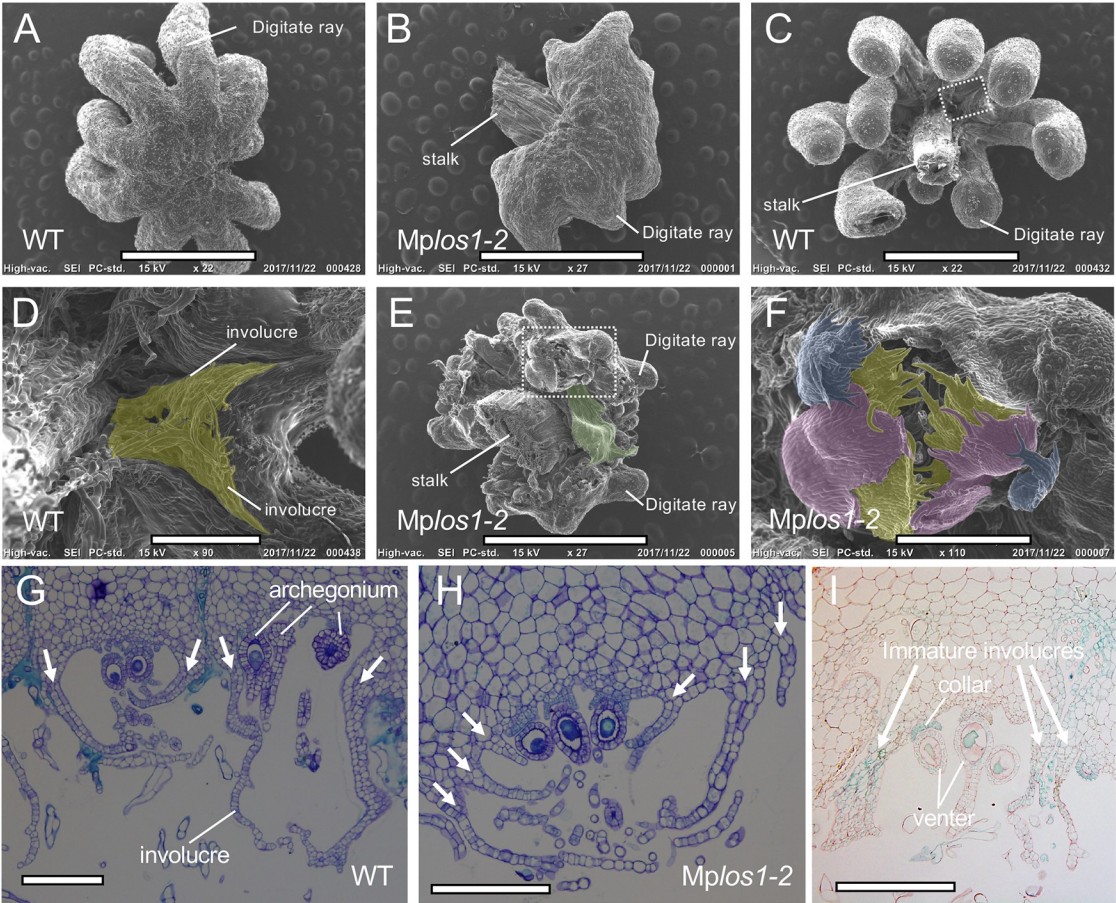

**Fig 5. MpLOS1 is necessary for the specification of lateral organ identity during reproductive growth.** (A and B) SEM images of the dorsal side of archegoniophores in WT Tak2 (A) and in Mp*los1-2* mutants (B). Note that archegoniophores in Mp*los1-2* mutants display shorter, malformed finger-like structures in place of digitate rays. (C-F) SEM images of the ventral side of archegoniophores in WT (C and D) and in Mp*los1-2* mutants (E and F). (D) and (F) are magnified images of (C) and (E), respectively. Instead of a single pair of involucres as developed by the WT (C and D), three pairs of membranous structures developed in Mp*los1-2* mutants (E and F). Enlarged leaf-like structures (highlighted in green) also developed in Mp*los1-2* mutants (E). Involucres in WT and the three pairs of membranous structures in Mp*los1-2* mutants are highlighted in yellow, red, or blue. (G and H) Cross sections of archegonial receptacles at regions between the digitate rays in WT (G) and in Mp*los1-2* mutants (H). Note the three pairs of ventral scalelike membranous structures in Mp*los1-2* mutants. Arrows indicate involucres or ventral scalelike structures. (I) Cross sections of GUS-stained archegoniophores that express *pro*Mp*LOS1:GUS*. GUS activity was detected in immature involucres. Scale bars = 2 mm in (A, B, C, and E), 400 μm in (D and F), and 200 μm in (G, H, and I). GUS, ß-glucuronidase; MpLOS1, *M. polymorpha* LOS1; *pro*Mp*LOS1*, promoter *M. polymorpha LOS1;* SEM, scanning electron microscope; Tak2, Takaragaike-2; WT, wild type.

sporophytic function in the lineage giving rise to rice and gametophytic functions in the lineage giving rise to liverworts when each originated the evolutionary novelty of lateral organs.

## Discussion

### Land plant ALOG proteins regulate lateral organ development and meristem activity

Here, we report the discovery that MpLOS1, a member of the ALOG protein family, controls both lateral organ development and apical meristem activity in *M. polymorpha*. MpLOS1 represses the growth of different lateral organs, including ventral scales and involucres, and Mp*LOS1* expression was detected early in the development of these lateral structures. These

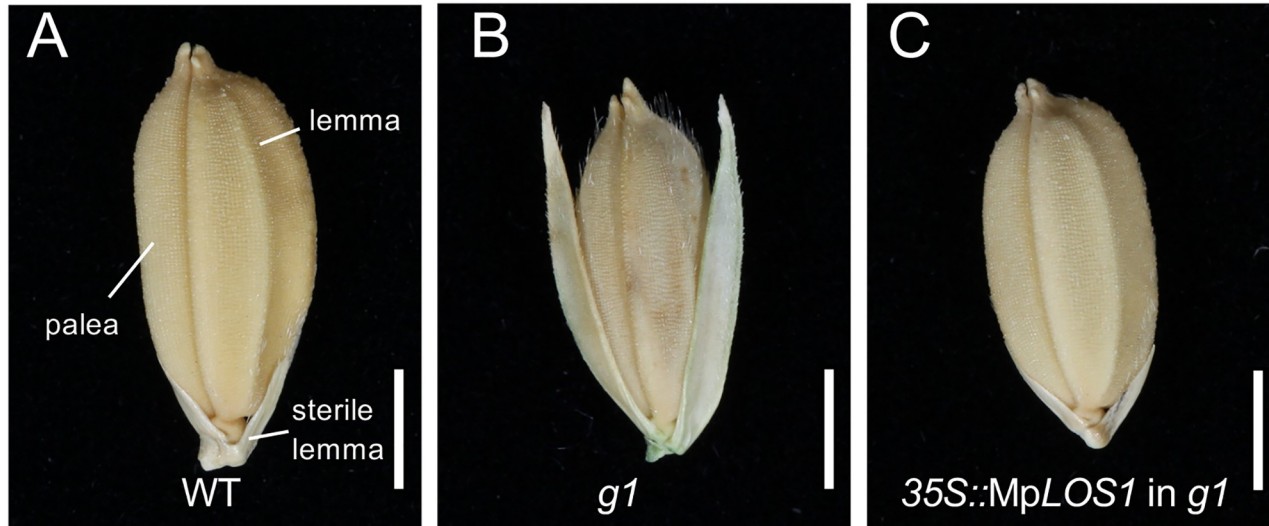

**Fig 6. Evolutionarily conserved ALOG family proteins in *Marchantia* and in rice specify analogous reduced lateral organs.** (A-C) Complementation of rice *alog* mutants with Mp*LOS1*. Phenotypes of a WT (A), a *g1* mutant (B), and a *g1* mutant expressing the *35Spro*::Mp*LOS1* construct (C) are shown. Scale bars = 2 mm in (A, B, and C). *35Spro, cauliflower mosaic virus 35S promoter*; ALOG, *Arabidopsis* LIGHT-DEPENDENT SHORT HYPOCOTYLS 1 and *Oryza* G1; *g1, long sterile lemma*; MpLOS1, *M. polymorpha* LATERAL ORGAN SUPRESSOR 1; WT, wild type.

data indicate that the gene is required for normal lateral organ development. Furthermore, MpLOS1 activity is required for apical meristem maintenance. However, Mp*LOS1* is not expressed in the apical meristems or surrounding cells. We propose that MpLOS1 cell autonomously regulates lateral organ development but non–cell autonomously regulates apical meristem maintenance (Fig 4I).

The role of ALOG proteins in meristem maintenance is conserved between monocots and dicots. *Oryza sativa* TAWAWA1(OsTAWAWA1) and *Solanum lycopersicum* TERMINATING FLOWER (SlTMF) (the tomato TAW1 homolog) proteins, members of the ALOG gene family in rice and tomato, respectively, repress maturation of meristems during reproductive growth [28,33,34]. Although the angiosperm genes control meristem development, neither SlTMF, *Arabidopsis thaliana* LSH3 (AtLSH3) (the *Arabidopsis* TAW1 homolog), nor OsTAW1 proteins are expressed in apical meristems. Instead, they are expressed at lateral organ boundaries [27,33,35]. Taken together, these data from a diversity of land plants suggest that although the ALOG genes act cell autonomously during the development of lateral organs, they act non–cell autonomously to control meristem development. It remains unclear how this might operate, but there is evidence from angiosperms that lateral organ development is required for meristem maintenance [9,36,37].

Taken together with our discovery that MpLOS1 is required non–cell autonomously for meristem maintenance in *M. polymorpha*, this means that the evolutionarily conserved ALOG family proteins control apical meristems in divergent plant lineages, in which the apical meristems are found in different phases of the life cycle. We propose that this mechanism for controlling shoot meristematic activity was already present in the last common ancestor of *Marchantia* and rice.

## Conserved ALOG proteins negatively regulate lateral organ growth

We discovered that MpLOS1 specifies lateral organ identity by negatively regulating the lateral organ outgrowth; involucres are transformed into ventral scalelike structures during

reproductive growth in Mp*los1-2* mutants (Fig 5G and 5H). The rice homolog, G1, also represses the development of lateral organs to specify the sterile lemmas. Loss-of-function mutations in *OsG1* result in the transformation of small sterile lemmas into large lemmas (Fig 6A and 6B) [29]. Similarly, in tomato, *Sltmf* mutants display a similar transformation, in which sepals develop leaf characteristics [33]. The conserved function of rice, tomato, and *M. polymorpha* LOS1 homologs suggests that the role of ALOG proteins in repression of lateral organ development is ancient. This functional conservation among divergent taxa of land plants suggests two alternative hypotheses regarding the evolution of lateral organs.

According to the first hypothesis, the common ancestor of liverworts and the seed plants developed lateral organs whose growth was controlled by one or more *ALOG* genes. Because lateral organs are present in the gametophyte and the sporophyte phase in liverworts and seed plants, respectively, under this scenario, the common ancestor may have had lateral organs in both the gametophyte and the sporophyte phase. These structures would then have subsequently diverged morphologically during the course of land plant evolution. However, this hypothesis is not well supported by the fossil record. The earliest polysporangiophyte and tracheophyte fossils we know of, such as *Aglaophyton* and *Cooksonia*, possessed simple axes devoid of lateral organs [20,21]. This suggests that their last common ancestor with the liverworts likewise lacked lateral organs.

An alternative hypothesis, which is in better accord with the fossil record, is therefore that the common ancestor of liverworts and the seed plants lacked lateral organs in the gametophyte and sporophyte phase but possessed an ALOG-mediated mechanism for controlling some other developmental process. The ALOG-dependent growth-repression mechanism was subsequently recruited independently during the evolution of lateral organs in the separate lineages leading to the liverworts and seed plants. If the last common ancestor did not develop lateral organs, and because ALOG function regulates lateral organ development in both liverworts and seed plants, we suggest that ALOG function was recruited independently during the evolution of lateral organs in different lineages of land plants. The original ALOG function prior to these independent recruitment events may have been to control some aspect of apical meristem activities because growth by apical meristems is a shared characteristic of land plants [38] and this ALOG function is found in both liverworts and seed plants. The recruitment of ALOG function during the independent evolution of lateral organs thus provides a molecular mechanism for the convergent evolution of growth repression in lateral organs.

## ALOG proteins may mediate diversification of lateral organs during plant evolution

It has been suggested that the repressive activity of the *OsG1* gene on growth led to the evolution of the rice spikelet [29]. Loss-of-function *Osg1* mutations revert the sterile lemma into a larger leafy structure, which has been interpreted as similar to a hypothetical ancestral structure (Fig 6A and 6B) [29]. According to this model, the formation of a pair of lower lemmas subtending the floret (rice flower surrounded by two bracts; the external lemma and internal palea) was the ancestral state. Then, during the evolution of rice, *OsG*1 activity was co-opted to repress the development of the lower lemma, which is now much reduced in size in modern rice compared with the ancestral state, resulting in the formation of the sterile lemma. It is formally possible that Mp*LOS1* may also have played a similar role in the evolution of lateral organs in liverworts. Several liverwort taxa with thalloid form are suggested to have evolved independently from ancestral leafy liverworts, in which leaves are hypothesized to be transformed into nonphotosynthetic ventral scales with reduced growth during this evolutionary transition [18,19]. We found that Mp*LOS1* is involved in the specification of lateral organ

identities by inhibiting cell division and chloroplast differentiation and that the loss of its function leads to the formation of chlorophyll-containing photosynthetic tissues (Fig 2J and 2K). These green appendages are in fact similar to the green-colored photosynthetic scales formed in the Treubiaceae family of liverworts, whose semithalloid form has been interpreted as an evolutionary transition state between the leafy and thalloid form [39–41]. Therefore, MpLOS1 function may also be associated with the evolution of the thallose body form by repressing leaf growth in ancestral leafy liverworts in the same way that OsG1 suppresses lower lemma development during rice spikelet evolution. It is possible that morphological modification of lateral organs is controlled by the spatial and temporal differences in expression levels of ALOG family genes, and this would provide a mechanism for the establishment of morphological diversification in lateral organs that develop on shoots during land plant evolution.

## Conclusion

We demonstrate that MpLOS1, a member of the ALOG gene family, plays a role in integrating meristem activity and lateral organ differentiation in *M. polymorpha*. MpLOS1 acts by repressing lateral organ growth and is required for meristem maintenance. Because ALOG proteins from angiosperms also repress lateral organ growth and are required for meristem maintenance, and these functions were rescued by MpLOS1, we conclude that protein functions of ALOG family proteins are conserved among these taxa and acted in their last common ancestor. We hypothesize that ALOG genes were co-opted to execute the morphological modification of analogous lateral organs during land plant evolution and contributed to the diversification of lateral organs in shoot systems during the course of land plant evolution.

## Materials and methods

### Plant materials and growth conditions

*M. polymorpha* Takaragaike-1 (Tak1, male) and Takaragaike-2 (Tak2, female) gemmalings were grown for 3–60 days at 22 ˚C under continuous light on petri plates containing 1/2 Gamborg's Basal Salt Mixture (B-5) growth medium (pH 5.5) with 1.2% agar (Nacalai tesque, Kyoto, Japan). Transition to reproductive growth was induced through far-red light supplementation [42].

### Generation of mutant plants

Knockout mutants of Mp*los1-1* (*vj99*) and Mp*los1-2* (*vj86*) were isolated by a mutant screen of spores from a cross between Tak1 and Tak2 transformed with the T-DNA vector pCambia1300 as previously described [15,43]. Knockout mutants of Mp*los1-3* and Mp*los1-4* were generated by gene-targeted homologous recombination [44]. Genetic nomenclature is outlined in [45].

### Phylogenetic analysis

Phylogenetic analysis was performed as described by Bowman and colleagues [3]. Protein sequences were collected using the *Marchantia* genome portal site MarpolBase (http://marchantia.info). Multiple sequence alignments were performed using the MUSCLE program [46] contained in the Geneious software (https://www.geneious.com). Gaps were removed by using Strip Alignment Columns in the Geneious package, and phylogenetic analyses were performed using PhyML (http://www.atgc-montpellier.fr/phyml/).

## Plasmid construction

For constructing the *pro*Mp*LOS1*:Mp*LOS1* plasmid that complements the Mp*los1* mutants, a Mp*LOS1* genomic fragment with a 10-kb upstream region and a 3-kb downstream region was amplified by PCR using Prime STAR GXL polymerase (TaKaRa, Shiga, Japan) and subcloned into pENTR/D-TOPO (Thermo Fisher Scientific, Waltham, Massachusetts, United States), which was subsequently integrated into pMpGWB101 by a Gateway LR reaction [47]. pENTR/D-TOPO that included the *pro*Mp*LOS1*:Mp*LOS1* complement fragment was modified to establish *pro*Mp*LOS1*:*eGFP*-Mp*LOS1* and *pro*Mp*LOS1*:*GUS* plasmids. A PCR-amplified *eGFP* coding sequence was inserted in frame with the 5′ end of the MpLOS1 coding sequence by the In-Fusion cloning reaction (TaKaRa, Shiga, Japan) to generate the *pro*Mp*LOS1*:*eGFP*-Mp*LOS1* plasmid. The coding sequence of MpLOS1 in the *pro*Mp*LOS1*:Mp*LOS1* complement fragment was replaced with a PCR-amplified GUS coding sequence by the In-Fusion cloning reaction to generate *pro*Mp*LOS1*:*GUS* plasmids. pENTR/D-TOPO vectors that included *pro*Mp*LOS1*:*eGFP*-Mp*LOS1* or *pro*Mp*LOS1*:*GUS* fragments were subsequently integrated into pMpGWB101 by the Gateway LR reaction. For overexpression of Mp*LOS1* in rice *g1* mutants, a plasmid carrying *pro35S*::Mp*LOS1* was constructed. The Mp*LOS1* genomic coding sequence was amplified by PCR and subcloned into pENTR/D-TOPO, which was subsequently integrated into pGWB2 by a Gateway LR reaction [48].

## Histochemical GUS staining

GUS staining was performed as described by Naramoto and colleagues [49], except that 50 mM sodium phosphate buffer was used. Samples were cleared with 70% ethanol and subsequently mounted using a clearing solution (chloral hydrate:glycerol:water, 8:1:2) for direct microscopic observation or dehydrated through a graded ethanol series and embedded in paraffin or Technovit 7100 resin for microtome sectioning.

## Plant embedding and sectioning

Plant material was fixed in FAA (45% ethanol:5% formaldehyde:5% acetic acid in water) for embedding in paraffin and Technovit 7100. For paraffin embedding, fixed plant material was dehydrated in a series of ethanol (25%–50%), t-butyl alcohol (10%–75%), and chloroform (20%) solutions and then embedded in Paraplast (McCormick, Baltimore, Maryland, USA). For Technovit 7100 embedding, fixed samples were dehydrated through a graded ethanol series and embedded in Technovit 7100 resin according to the manufacturer's instructions (Heraeus Kulzer, Hanau, Germany). Embedded samples were sectioned on a rotary microtome into a series of vertical transverse and longitudinal sections (thickness of 8 μm for paraffin and 4 μm for Technovit sectioning). The obtained sections were further treated with neutral red dyes as a counterstain for GUS-stained samples or toluidine blue for the other samples. Multi-Mount 480 solution (MATSUNAMI, Osaka, Japan) or Entellan new (MERCK, Darmstadt, Germany) were used as mounting agents to preserve the samples on the slides.

## ClearSee treatment and staining of cell walls

Plants were fixed with 4% paraformaldehyde (PFA) in 1× PBS for 1 hour at room temperature under vacuum. Samples were subsequently washed twice with PBS and transferred to ClearSee solution (10% xylitol, 15% sodium deoxycholate, and 25% urea in water) [50]. ClearSee treatment was prolonged until samples became transparent. Cell walls were stained for 1 hour with 0.1% (v/v) calcofluor or with 0.1% (w/v) Direct Red 23 dissolved in ClearSee. Stained samples were washed for at least 30 minutes with ClearSee solution before observation.

### EdU uptake experiments

Gemmalings were incubated in 1/2 B5 medium containing 10 μM EdU (Click-iT EdU Alexa Fluor 488 imaging kit; Thermo Fisher, Waltham, Massachusetts, USA) for 3 hours. Samples were fixed with 4% PFA in 1× PBS for 1 hour under vacuum and then washed three times in PBS. Coupling of EdU to the Alexa Fluor substrate was performed according to the manufacturer's instructions. Before observations, samples were cleared with ClearSee solution, and cell walls were subsequently stained by Direct Red 23.

### Microscopy

Anatomical features were observed with a light microscope (Olympus BX51) equipped with an Olympus DP71, a light sheet microscope (Zeiss Z.1), or a confocal laser scanning microscope (Olympus FV1000 or Zeiss LSM880). For light microscope observations, a PLAPON 2× objective, a UPlanFl 10× objective, or a UPlanFl 20× objective was used. Light sheet microscope observations were conducted using Lightsheet Z.1 detection optics 5× or Clr Plan-Neofluar 20×. For confocal laser scanning microscopy, cell walls stained by Calcofluor or by Direct Red 23 were excited at 405 nm or 543 nm, respectively, whereas GFP and Alexa 488–labeled EdU were excited at 488 nm. Samples were mounted using ClearSee solution and observed with silicon oil objectives. The 3D reconstruction was done by using Imaris software (BITPLANE, http://www.bitplane.com/). High-resolution images showing ultrastructural details were obtained using an SEM (JEOL JCM-6000Plus NeoScope) and an FESEM (Hitachi SU820).

### Data analysis and statistics

Four alleles of the Mp*los1* mutants, including Mp*los1-1*, Mp*os1-2*, Mp*los1-3*, and Mp*los1-4* displayed identical phenotypes in vegetative growth and, thus, Mp*los1-1* mutants were used as a representative allele unless otherwise described. All experiments were repeated at least three times. Representative images were used for the preparation of figures. Statistical analysis was conducted using Excel. For comparing two groups, we used the *t* test to calculate the *p*-value. The *p*-values of the relevant experiments are described in the Supporting information (S2 Data). Materials Design Analysis Reporting checklist is described in the Supporting information (S1 MDAR Checklist).

### Supporting information

**S1 Fig. Responsible gene of *vj99* mutant encodes an ALOG family protein (related to Fig 1).** (A and B) Phenotypes of *vj99* and *vj86* mutants. (C and D) LSFM image of gross morphology of WT (C) and *vj99* mutant (D) gemmalings. (E) Overview of the functional Mp*LOS1* construct and the T-DNA insertion mutants isolated by forward genetic screening. The regions 10,152 bp upstream and 1,689 bp downstream of coding sequences were used to express Mp*LOS1*. (F and G) Phenotypic series of Mp*los1* knockout mutants. (H-K) Complementation of Mp*los1-1* mutants with a functional Mp*LOS1* construct. The phenotype of Mp*los1-1* (I) is complemented by introducing the genomic Mp*LOS1* fragment (J) as well as the *eGFP*-fused Mp*LOS1* genomic fragment (K). (L) Phylogenetic tree of ALOG family proteins in *Arabidopsis*, rice, and *Marchantia*. Green, red, and blue symbols indicate ALOG proteins in *Arabidopsis*, rice, and *Marchantia*, respectively. Alignment of ALOG family proteins can be found in S1 Data. Scale bars = 1 cm in (A, B, F, G, H, I, J, and K) and 500 μm in (C and D). ALOG, *Arabidopsis* LIGHT-DEPENDENT SHORT HYPOCOTYLS 1 and

*Oryza* G1; eGFP, enhanced green fluorescent protein; LSFM, light sheet fluorescence microscopy; MpLOS1, *M. polymorpha* LATERAL ORGAN SUPRESSOR 1; T-DNA, transfer DNA; WT, wild type.
(TIF)

**S2 Fig. The mutation in Mp*LOS1* transforms ventral scales into chlorophyll-containing photosynthetic tissues with an increased number of cells (related to Fig 2).** (A and B) LSFM image of gemmalings in WT (A) and in Mp*los1-1* mutants (B) observed from the ventral side. Dotted boxes in (A) and (B) that include apical notch regions are shown as close-up images in Fig 2(E) and 2(F), respectively. Ventral scales in WT or their corresponding tissues in Mp*los1-1* mutants are indicated by arrows. (C-F) Vertical transverse optical sections of apical notch regions in 4-day-old gemmalings obtained by LSFM. Optical sections in WT (C and D) and in Mp*los1-1* mutants (E and F) are shown. Note that the number of cells that make up mucilage and ventral scales increased, and thus, these tissues became larger in Mp*los1-1* mutants. (G and H) Images of ventral scales in WT (G) and the corresponding tissues in Mp*los1-1* mutants (H). Note that ventral scale cells are transformed into green tissues that lack rhizoids and mucilage hair cells in Mp*los1-1* mutants. Rhizoids and mucilage hair in WT are indicated by an arrowhead and an arrow, respectively. (I and J) FESEM images of ventral scale cells in WT (I) and the corresponding cells in Mp*los1-1* mutants (J). Images were composites, made up of 25 and 30 individual images (tiles) in WT and Mp*los1-1* mutants, respectively. Scale bars = 150 μm in (A and B), 100 μm in (C and E), 200 μm in (G and H), and 10 μm in (I and G). MpLOS1, *Marchania polymorpha* LATERAL ORGAN SUPRESSOR 1; LSFM, light sheet fluorescence microscopy; FESEM, field emission scanning electron microscopy; WT, wild type.
(TIF)

**S3 Fig. MpLOS1 plays crucial roles in bifurcation (related to Fig 3).** (A and B) Apices stained by *pro*Mp*YUC2:GUS* in 3-week-old gemmalings in WT (A) and Mp*los1-1* mutants (B). Arrowheads indicate GUS staining at apical notches. (C) Number of gemma cups in WT Tak1 and in Mp*los1-1* mutants. Each bar indicates the mean ± SD. At least seven gemmalings were analyzed at each time point. (D and E) EdU labeling in central lobes. EdU-positive signals of 5-day-old WT gemmalings (D) and that of 9-day-old WT gemmalings (E) are shown. Note that there is little EdU labeling in the central lobes in 9-day-old gemmalings. Arrowhead indicates central lobes. Scale bars = 1.5 mm in (A and B) and 200 μm in (D and E). *p*-Values lower than 0.01 are indicated by asterisks (*). Underlying data for this figure can be found in S2 Data. EdU, 5-ethynyl-2′-deoxyuridine; GUS, *ß-glucuronidase*; MpLOS1, *M. polymorpha* LATERAL ORGAN SUPRESSOR 1; *pro*Mp*YUC2*, promoter *M. polymorpha* YUCCA2; Tak1, Takaragaike-1; WT, wild type.
(TIF)

**S4 Fig. MpLOS1 plays crucial roles in maintenance of meristem activities (related to Fig 3).** (A-H) Detailed cellular organization of apical notches in 3-day-old gemmalings in WT (A, C, E, and G) and in Mp*los1-1* mutants (B, D, F, and H). Horizontal (A and B), vertical transversal (C and D), and vertical longitudinal optical sections (E-H) obtained after the 3D reconstruction of a series of CLSM images were shown. The cells highlighted by orange or cyan in (A) and (B) were sectioned in vertical longitudinal and vertical transversal directions, respectively. The cells highlighted by orange or cyan in (A-H) are identical cells. Cell walls were stained using Direct Red 23. (I) SEM image of Mp*los1-1* mutant gemmalings. Thallus regeneration occurs in Mp*los1-1* mutants next to aborted meristems. The aborted meristems are indicated by dotted boxes. Scale bars = 30 μm in (A-H) and 1 mm in (I). CLSM, confocal laser

scanning microscopy; MpLOS1, *M. polymorpha* LATERAL ORGAN SUPRESSOR 1; SEM, scanning electron microscope; WT, wild type.
(TIF)

**S5 Fig. Functional eGFP-MpLOS1 proteins were detected in ventral scales and the cells beneath the basal part of the ventral scales (related to Fig 4).** CLSM image of Mp*los1-1* mutant gemmalings that express *pro*Mp*LOS1:eGFP-*Mp*LOS1* constructs. Vertical transverse sections were obtained after the 3D reconstruction of a series of CLSM images. Cell walls were stained by calcofluor. Scale bar = 50 μm. CLSM, confocal laser scanning microscopy; eGFP, enhanced green fluorescent protein; MpLOS1, *M. polymorpha* LATERAL ORGAN SUPRESSOR 1.
(TIF)

**S6 Fig. MpLOS1 plays crucial roles in lateral organ differentiation in reproductive growth (related to Fig 5).** (A-F) SEM image of antheridiophores in WT Tak1 (A and D) and Mp*los1-1* mutants (B, C, E, and F). The dorsal side (A-C) and ventral side (D-F) of antheridiophores are shown. (C) and (F) are close-up images of (B) and (E), respectively. Some ventral scales are highlighted by colors. Note the exaggerated growth of ventral scales as compared with the size of thalli (F). (G and H) Vertical sections of antheridia in WT (G) and Mp*los1-1* mutants (H). Note the extra cell division of misspecified ventral scales in Mp*los1-1* mutants. Ventral scales (G) and misspecified ventral scales (H) are indicated by arrows. (I and J) Cross sections of GUS-stained antheridiophores (I) and antheridia (J) that express *pro*Mp*LOS1:GUS* constructs. Note that GUS activities were detected in the ventral scales in WT (I). (K and L) Cross sections of GUS-stained archegoniophores that express *pro*Mp*LOS1:GUS*. Regions that include mature involucres (K) and the whole image of archegoniophores (L) are shown. Note that in contrast to immature archegoniophores, GUS activity was not detected in mature involucres. Scale bars = 2 mm in (A, B, and E), 1 mm in (F), 400 μm in (C), and 200 μm in (G, H, I, K, and L). GUS, *ß-glucuronidase*; MpLOS1, *M. polymorpha* LATERAL ORGAN SUPRESSOR 1; SEM, scanning electron microscope; Tak1, Takaragaike-1; WT, wild type.
(TIF)

**S1 Movie. The 3D reconstruction of the apical notch region by using CLSM data that display eGFP-MpLOS1 and cell walls.** Cell walls were stained by calcofluor. CLSM, confocal laser scanning microscopy; eGFP, enhanced green fluorescent protein; MpLOS1, *M. polymorpha* LATERAL ORGAN SUPRESSOR 1.
(MP4)

**S1 Data. Alignment of ALOG family proteins data under fasta format.** ALOG, *Arabidopsis* LIGHT-DEPENDENT SHORT HYPOCOTYLS 1 and *Oryza* G1.
(FASTA)

**S2 Data. Numerical data related to this manuscript.**
(XLSX)

**S1 MDAR Checklist. MDAR checklist file related to this manuscript.** MDAR, Materials Design Analysis Reporting.
(DOCX)

## Acknowledgments

We thank Kei Saito, Eriko Kida, and Kanane Sato for assistance with transformation and microtome sectioning. We also thank Mayumi Wakazaki for preparing samples for electron microscopy.

## Author Contributions

**Conceptualization:** Satoshi Naramoto, Victor Arnold Shivas Jones, Masaki Shimamura, Liam Dolan.

**Data curation:** Satoshi Naramoto.

**Formal analysis:** Satoshi Naramoto.

**Funding acquisition:** Satoshi Naramoto, Junko Kyozuka.

**Investigation:** Satoshi Naramoto, Victor Arnold Shivas Jones, Nicola Trozzi, Mayuko Sato, Kiminori Toyooka, Masaki Shimamura, Sakiko Ishida, Kazuhiko Nishitani, Kimitsune Ishizaki, Ryuichi Nishihama, Takayuki Kohchi, Liam Dolan.

**Methodology:** Satoshi Naramoto.

**Project administration:** Satoshi Naramoto.

**Resources:** Satoshi Naramoto, Victor Arnold Shivas Jones, Kimitsune Ishizaki, Ryuichi Nishihama, Takayuki Kohchi, Liam Dolan.

**Supervision:** Satoshi Naramoto.

**Validation:** Satoshi Naramoto.

**Visualization:** Satoshi Naramoto.

**Writing – original draft:** Satoshi Naramoto, Victor Arnold Shivas Jones, Masaki Shimamura, Liam Dolan.

**Writing – review & editing:** Satoshi Naramoto, Victor Arnold Shivas Jones, Liam Dolan, Junko Kyozuka.

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
