## [Editor Report · Decision Letter 0]

12 Jul 2019

Dear Dr Naramoto, 

Thank you for submitting your manuscript entitled "A conserved regulatory mechanism mediates the convergent evolution of plant shoot lateral organs" for consideration as a Research Article by PLOS Biology.

Your manuscript has now been evaluated by the PLOS Biology editorial staff, as well as by an academic editor with relevant expertise, and I'm writing to let you know that we would like to send your submission out for external peer review.

**Important**: Please also see below for further information regarding completing the MDAR reporting checklist. The checklist can be accessed here: https://plos.io/MDARChecklist

Please re-submit your manuscript and the checklist, within two working days, i.e. by Jul 16 2019 11:59PM.

Kind regards,

Roli Roberts

Senior Editor

PLOS Biology

INFORMATION REGARDING THE REPORTING CHECKLIST:

PLOS Biology is pleased to support the "minimum reporting standards in the life sciences" initiative (https://osf.io/preprints/metaarxiv/9sm4x/). This effort brings together a number of leading journals and reproducibility experts to develop minimum expectations for reporting information about Materials (including data and code), Design, Analysis and Reporting (MDAR) in published papers. We believe broad alignment on these standards will be to the benefit of authors, reviewers, journals and the wider research community and will help drive better practise in publishing reproducible research. 

We are therefore participating in a community pilot involving a small number of life science journals to test the MDAR checklist. The checklist is intended to help authors, reviewers and editors adopt and implement the minimum reporting framework. 

IMPORTANT: We have chosen your manuscript to participate in this trial. The relevant documents can be located here:

MDAR reporting checklist (to be filled in by you): https://plos.io/MDARChecklist

**We strongly encourage you to complete the MDAR reporting checklist and return it to us with your full submission, as described above. We would also be very grateful if you could complete this author survey:

https://forms.gle/seEgCrDtM6GLKFGQA

Additional background information:

Interpreting the MDAR Framework: https://plos.io/MDARFramework

Please note that your completed checklist and survey will be shared with the minimum reporting standards working group. However, the working group will not be provided with access to the manuscript or any other confidential information including author identities, manuscript titles or abstracts. Feedback from this process will be used to consider next steps, which might include revisions to the content of the checklist. Data and materials from this initial trial will be publicly shared in September 2019. Data will only be provided in aggregate form and will not be parsed by individual article or by journal, so as to respect the confidentiality of responses. 

Please treat the checklist and elaboration as confidential as public release is planned for September 2019.

We would be grateful for any feedback you may have.

---

## [Decision Letter · Decision Letter 1]

16 Aug 2019

Dear Dr Naramoto,

Thank you very much for submitting your manuscript "A conserved regulatory mechanism mediates the convergent evolution of plant shoot lateral organs" for consideration as a Research Article at PLOS Biology. Your manuscript has been evaluated by the PLOS Biology editors, an Academic Editor with relevant expertise, and by two independent reviewers.

You'll see that both of the reviewers are broadly positive about your study, but each raises a number of concerns that should be addressed before we can consider the manuscript further. IMPORTANT: After discussion with the AE, we will not insist on the experiment requested by reviewer #1 in his point 5 (rescue of the rice mutant by MpTAW1) - although this might further strengthen the paper, we believe that it is beyond the scope of the current study.

In light of the reviews (below), we will not be able to accept the current version of the manuscript, but we would welcome resubmission of a much-revised version that takes into account the reviewers' comments. We cannot make any decision about publication until we have seen the revised manuscript and your response to the reviewers' comments. Your revised manuscript is also likely to be sent for further evaluation by the reviewers.

Your revisions should address the specific points made by each reviewer. Please submit a file detailing your responses to the editorial requests and a point-by-point response to all of the reviewers' comments that indicates the changes you have made to the manuscript. In addition to a clean copy of the manuscript, please upload a 'track-changes' version of your manuscript that specifies the edits made. This should be uploaded as a "Related" file type. You should also cite any additional relevant literature that has been published since the original submission and mention any additional citations in your response. 

Before you revise your manuscript, please review the following PLOS policy and formatting requirements checklist PDF: http://journals.plos.org/plosbiology/s/file?id=9411/plos-biology-formatting-checklist.pdf. It is helpful if you format your revision according to our requirements - should your paper subsequently be accepted, this will save time at the acceptance stage.

Please note that as a condition of publication PLOS' data policy (http://journals.plos.org/plosbiology/s/data-availability) requires that you make available all data used to draw the conclusions arrived at in your manuscript. If you have not already done so, you must include any data used in your manuscript either in appropriate repositories, within the body of the manuscript, or as supporting information (N.B. this includes any numerical values that were used to generate graphs, histograms etc.). For an example see here: http://www.plosbiology.org/article/info%3Adoi%2F10.1371%2Fjournal.pbio.1001908#s5.

For manuscripts submitted on or after 1st July 2019, we require the original, uncropped and minimally adjusted images supporting all blot and gel results reported in an article's figures or Supporting Information files. We will require these files before a manuscript can be accepted so please prepare them now, if you have not already uploaded them. Please carefully read our guidelines for how to prepare and upload this data: https://journals.plos.org/plosbiology/s/figures#loc-blot-and-gel-reporting-requirements.

Upon resubmission, the editors will assess your revision and if the editors and Academic Editor feel that the revised manuscript remains appropriate for the journal, we will send the manuscript for re-review. We aim to consult the same Academic Editor and reviewers for revised manuscripts but may consult others if needed.

We expect to receive your revised manuscript within two months. Please email us (plosbiology@plos.org) to discuss this if you have any questions or concerns, or would like to request an extension. At this stage, your manuscript remains formally under active consideration at our journal; please notify us by email if you do not wish to submit a revision and instead wish to pursue publication elsewhere, so that we may end consideration of the manuscript at PLOS Biology.

When you are ready to submit a revised version of your manuscript, please go to https://www.editorialmanager.com/pbiology/ and log in as an Author. Click the link labelled 'Submissions Needing Revision' where you will find your submission record. 

Sincerely,

Roli Roberts

Senior Editor

PLOS Biology

REVIEWERS' COMMENTS:

Reviewer #1:

[identifies himself as Hiro-Yuki Hirano]

Naramoto et al. revealed that the MpTAWAWA1 encoding an ALOG family protein is required for the development of lateral organs in Marchantia polymorpha, and that the function of ALOG genes to repress lateral organ growth is conserved in bryophytes and angiosperms. The authors identified the gene responsible for the morphological defects such as the formation of abnormal outgrowth and hyponasty of the thallus; the gene encoded a protein belonging to the ALOG family, some members of which are involved in organ and inflorescence development in rice. From the close examination of mutant phenotypes, the authors revealed that MpTAW1 is likely to regulate the development of lateral organs, such as ventral scales in the thallus and involucres in archegoniophore and to maintain the activity of the apical meristem in M. polymorph. Because MpTAW1 was expressed in the lateral organ primordia but not in the meristem, the authors concluded that this gene acts cell autonomously on the lateral organ development and non-cell autonomously on meristem activity. In addition, the authors showed that MpTAW1 rescued a mutant phenotype of the rice g1 mutant, suggesting that protein function of ALOG members is conserved in bryophyte and angiosperms. Thus, ALOG genes regulate the lateral organ development in the gametophyte in the bryophytes and in the sporophyte in the angiosperms. 

 The results and the interpretation are interesting, and this work is likely to contribute the understanding of not only Marchantia development but also Evo-devo studies in plants. However, several concerns are raised, and I hope they are helpful to improve the work. 

Major point:

1. ‘MpTAW1 activity is required for….’, line 142-199 

I am confused by the description of the results and interpretation in this section. The authors first described that the expansion of the region between the two apical notches (central lobe) is defective, resulting in the high density of the gemma cups. Then the authors examined the incorporation of EdU in the apical meristem to check the apical notch separation, that is, the expansion of the central lobe. 

I am wondering if the lobe expansion is a direct consequence of the activity of the apical meristem. The lobe is outside of the apical meristem consisting of the apical cell and merophytes. Thus, I think that the cells in the central lobe probably proliferate independent of the apical meristem activity. In Arabidopsis root, cells divide and proliferate in the division zone for root growth after differentiation of each cells such as epidermal and cortex cells originated from the root apical meristem. I think that the cell proliferation in the central lobe corresponds to that in the cell division zone of Arabidopsis root. 

Therefore, to uncover the cause of a reduction in central lobe growth in the Mptaw1 mutant, the authors should examine whether cell division is impaired in the central lobes during its growth. Conversely, if the cells in the central lobes are supplied only from the apical meristem, such data should be shown. 

2. Apart from the above comment, I agree that MpTAW1 is required for the activity of the apical meristem, as the authors showed lower incorporation of EdU. An additional examination such as the observation of division pattern of the apical cell in Mptaw1 would bring information that strengthen the authors’ statement in which MpTAW1 is required for the activity of apical meristem. 

3. line 248-250

The authors stated that ‘loss of MpTAW1 function results in the homeotic transformation of involucres into more scale-like structures’, and ‘homeotic transformation’ is repeatedly used in DISCUSSION. However, there is little evidence that the abnormal involucres in the Mptaw1 mutant has ventral scale identity. To assert homeotic transformation, more close morphological examination should be added. Otherwise, I suggest that the authors weaken their claim by removing the word ‘homeotic transformation’ . 　 

4. The usage of the word ‘axis’

I am confused by the word ‘axis’. First, what does ‘axial systems’ (line 50, 52) mean? 

Second, it is also unclear the following words: ‘a bifurcating axis (line 143)’, ‘axis separation (line 148)’, ‘axis elongation (line 150)’, and ‘axis development (line 155)’. The authors should define these words more clearly. Some of the ‘axes’ should be drawn in the Figure 3A, if possible. But I think that the authors can describe this section more clearly (easily to understand) without the use of ‘axis’. 

5. The TAW1, a rice ALOG gene, controls the activity of the inflorescence meristem. The authors claim that MpTAW1 is associated with the apical meristem activity. So, I would like to know whether MpTAW1 rescue the phenotype of loss-of-function mutant of TAW1. 

6. The ALOG genes in rice were already named in a preceding paper (Yoshida et al., 2009). But the authors used the different names such as TAW2 and TAW3 in Figure S1. The change of the gene names should be avoided, because it brings confusion. The name of MpTAW1 sound a little bit strange, because the function of MpTAW1 is similar to rice G1 rather than rice TAW1. If the authors want to avoid the use of MpG1, it would be good to create a new name from the mutant phenotype because this study started with the identification of morphological mutant. 

7. Transformation of rice g1 mutant.

What promoter did the authors use to express MpTAW1 in rice? There is no description about the experimental procedure concerning rice transformation in MATERIALS AND METHODS.

Are any effects other than the repression of sterile lemma growth observed in the transgenic g1 mutant overexpressing MpTAW1? Why the original lemma was not affected by MpTAW1 overexpression? 

Minor points:

1. RESULTS should be written in past tense. 

2. line 136: ‘a higher photosynthetic activity in this tissue’

Is it possible to lead this discussion from only starch accumulation?

3. ‘Molecular function of ALOG proteins (line 258, 267, 373)’ should be ‘protein function of ALOGs’, because there is no molecular analysis in the experiment using rice. 

4. line 312, ‘The rice homolog…the sterile lemmas’ 

The reference should be cited. 

5. Figure 2E: 

The organ with arrows appears to be out of focus a little. 

6. Figures 4A-C:

Are these images taken from the ventral side? If so, please indicate this.

Reviewer #2:

[also please see attached word doc for formatted version]

This paper presents interesting and significant results on the regulation of lateral appendage development. These results have important implications for our understanding of the evolution of lateral appendages.

The paper is well written and illustrated. The methods and experimental design are appropriate and solid.

I see a minor issue with the statement on thylakoids (line 134): How many chloroplasts did you assess for that, i.e., what kind of numerical data support this statement about thylakoids?

The only other qualm I have, is with the section of the discussion on lines 320-324, concerning the two alternative hypotheses advanced by the authors regarding the evolution of lateral organs. Your first hypothesis is not specific enough and I don’t see how it would work. Here is why: if the ALOG gene controlled the development of lateral organs in the last common ancestor of the liverworts and the seed plants, where did this common ancestor have lateral organs: in the gametophyte, in the sporophyte, in both phases? The latter should have been the case, given that liverworts have lateral appendages in the gametophyte and seed plants have lateral appendages in the sporophyte. In this case we are, consequently, talking about a plant that had lateral appendages in both the gametophyte and the sporophyte phase and was the last common ancestor of liverworts and seed plants. This ancestor is the same as the last common ancestor of liverworts and tracheophytes (or polysporangiophytes). We don’t know what the earliest liverworts looked like, but it is likely that their gametophytes were thalloid (as suggested by their oldest known fossils) and could have had scales as seen in Marchantia. However, the oldest tracheophytes (and polysporangiophytes) had simple sporophyte axes devoid of lateral appendages. On the other hand, according to at least some phylogenies regarding relationships at the base of embryophytes (if not most of those phylogenies), the two ancestors you are talking about in the two hypotheses could have been the same. So, I think the two hypotheses need to be thoroughly reassessed and this part of the discussion needs to be re-written. Maybe the alternative hypotheses resulting from such a reassessment will have to do with whether the common ancestor had lateral appendages or not and in which phase of the life cycle (gametophyte or sporophyte) it had appendages. Furthermore, if ALOG function in lateral organ development was recruited independently in different embryophyte (land plant) lineages (lines 333-334), you may want to consider and speculate on what would have been their shared function (i.e. in some common ancestor) prior to these independent recruitment events.

Other than that, I only have a few terminology, grammar, and wording suggestions:

- Throughout the manuscript: check the grammar – I found some inadvertent plurals

- Line 31 and throughout – Marchantia is a formal genus name and it should italicized

- Lines 37-38 – the wording is a bit counterintuitive if you say that they specify the identity of lateral organs by repressing their growth. Please reword.

- Lines 59-60 – … forms a gametophyte axis that …

- Lines 69-70 – … and bilateral symmetry of the Marchantia scales resemble …

- Line 74 – is this, rather, Kenrick & Crane 1997? Does it not need to be a numbered reference?

- Line 116 – … much shorter than in the WT

- Lines 145-146 – “distance between each apical notch”??

- Lines 147-148, duplication of apical notches and axis separation – why not refer to this as bifurcation or, even better, just simply branching? After all, it is the same type of process as seen in the apical branching of tracheophyte sporophyte axes.

- Line 186 – the term merophyte has a more specific meaning than the general “cells derived from the apical cell”. The cells produced directly by division of the apical cell are designated as apical derivatives. A merophyte is a group of clonally related cells resulting from sequential cell divisions that originate in a single derivative of the apical cell of a meristem.

- Line 188 and elsewhere – triangular is a misleading descriptor of the shape of these cells, at least as illustrated in the figures referenced. Wedge-shaped may be a more accurate term.

- Line 253 – I have never encountered the term collar in describing archegonium parts. Do you mean archegonial neck?

- Lines 260-261 – your wording implies that a spikelet is a unit of a flower, which is inaccurate. Instead, the spikelet is made of one to several flowers. Please reword.

- Line 302 – your wording implies that rice and Marchantia are the earliest land plants. Please reword.

- Line 314 – Similarly, in tomato, Sltmf mutants …

- Line 374 – check wording

---

## [Decision Letter · Decision Letter 2]

25 Oct 2019

Dear Dr Naramoto,

Thank you for submitting your revised Research Article entitled "A conserved regulatory mechanism mediates the convergent evolution of plant shoot lateral organs" for publication in PLOS Biology. I have now obtained advice from the original reviewers and have discussed their comments with the Academic Editor. 

Based on the reviews, we will probably accept this manuscript for publication, assuming that you will modify the manuscript to address the remaining points raised by reviewers #1. IMPORTANT: Please also make sure to address the Data Policy and other policy-related requests noted at the end of this email.

We expect to receive your revised manuscript within two weeks. Your revisions should address the specific points made by each reviewer. In addition to the remaining revisions and before we will be able to formally accept your manuscript and consider it "in press", we also need to ensure that your article conforms to our guidelines. A member of our team will be in touch shortly with a set of requests. As we can't proceed until these requirements are met, your swift response will help prevent delays to publication.

*Copyediting*

*Published Peer Review History*

*Early Version*

*Submitting Your Revision*

Sincerely,

Roli Roberts

Senior Editor

PLOS Biology

DATA POLICY:

Regardless of the method selected, please ensure that you provide the individual numerical values that underlie the summary data displayed in the following figure panels as they are essential for readers to assess your analysis and to reproduce it: Figs 3DG, S3C, and the alignment for the tree shown in Fig S1L. NOTE: the numerical data provided should include all replicates AND the way in which the plotted mean and errors were derived (it should not present only the mean/average values).

REVIEWERS' COMMENTS:

Reviewer #1:

[identifies himself as Hiro-Yuki Hirano]

The authors have addressed most of my concerns and revised the manuscript.

Before publication, however, the authors should check more carefully the manuscript to remove inappropriate expression and grammatical errors. In addition, consistency should be maintained throughout the manuscript; eg., the usage of the mutant name as described below. I suggest the manuscript should be checked by a native speaker of English who preferably understand biology. Probably due to the error when the manuscript was converted to PDF file, figure legends are inserted inappropriate positions in the text. 

- There are a number of inconsistent usages of mutant names: los1 mutant, los1-1 mutant, or los1 mutants, including the presence or absence of definite article (the). If the authors used one allele of los1 for phenotypic analysis, the name should be used consistently. Otherwise, respective allele names should be used.

Reviewer #2:

[identifies himself as Alexandru M.F. Tomescu]

I am happy with the revisions you have included.

---

## [Editor Report · Decision Letter 3]

11 Nov 2019

Dear Dr Naramoto,

On behalf of my colleagues and the Academic Editor, Mark Estelle, I am pleased to inform you that we will be delighted to publish your Research Article in PLOS Biology. 

Early Version

PRESS 

Kind regards,

Alice Musson

Publications Assistant, 

PLOS Biology

on behalf of

Roland Roberts,

Senior Editor

PLOS Biology